# BasisNet: Two-stage Model Synthesis for Efficient Inference

## Abstract

We present BasisNet which combines recent advancements in efficient neural network architectures, conditional computation, and early termination in a simple new form. Our approach uses a lightweight model to preview an image and generate input-dependent combination coefficients, which are later used to control the synthesis of a specialist model for making more accurate final prediction. The two-stage model synthesis strategy can be used with any network architectures and both stages can be jointly trained end to end. We validated BasisNet on ImageNet classification with MobileNets as backbone, and demonstrated clear advantage on accuracy-efficiency trade-off over strong baselines such as EfficientNet (Tan & Le, 2019), FBNetV3 (Dai et al., 2020) and OFA (Cai et al., 2019). Specifically, BasisNet-MobileNetV3 obtained 80.3% top-1 accuracy with only 290M Multiply-Add operations (MAdds), halving the computational cost of previous state-of-the-art without sacrificing accuracy. Besides, since the first-stage lightweight model can independently make predictions, inference can be terminated early if the prediction is sufficiently confident. With early termination, the average cost can be further reduced to 198M MAdds while maintaining accuracy of 80.0%.

## 1 Introduction

High-accuracy yet low-latency convolutional neural networks enable opportunities for on-device machine learning, and are playing increasingly important roles in various mobile applications, including but not limited to intelligent personal assistants, AR/VR and real-time voice translations. Designing efficient convolutional neural networks especially for edge devices has received significant research attention. Prior research attempted to tackle this challenge from different perspectives, such as novel network architectures (Howard et al., 2017; Sandler et al., 2018; Ma et al., 2018; Zhang et al., 2018; Howard et al., 2019), better incorporation with hardware accelerators (Lee et al., 2019), or conditional computation and adaptive inference algorithms (Bolukbasi et al., 2017; Figurnov et al., 2017; Leroux et al., 2017; Wang et al., 2018; Marquez et al., 2018). However, focusing on one perspective *in isolation* may have side effects. For example, novel network architectures may introduce custom operators that are not well-supported by hardware accelerators, thus a promising new model may have limited practical improvements on real devices due to a lack of hardware support. We believe that these three perspectives should be better integrated to form a more holistic general approach that ensures the broader applicability of the resulting system.

In this paper, we present BasisNet, which takes advantage of progress in all these perspectives and combines several key ideas in a simple new form. The core idea behind BasisNet is *dynamic model synthesis*, which aims at efficiently generating sample-dependent specialist model from a collection of bases, so the resultant model is specialized at handling the given input and can give more accurate predictions. This concept is flexible and can be applied to any *novel network architectures*. On the *hardware side*, the two-stage model synthesis strategy allows the execution of the lightweight and synthesized specialist model on different processing units (e.g., CPU, mobile GPUs, dedicated accelerators, etc.) in parallel to better handle streaming data. The BasisNet design is naturally compatible with *early termination*, and can easily balance between computation budget and accuracy with a single hyperparameter (prediction confidence).

An overview of the BasisNet is shown in Fig. 1. Using image classification as an example, our BasisNet has two stages: the first stage relies on a *lightweight model* to preview the input image

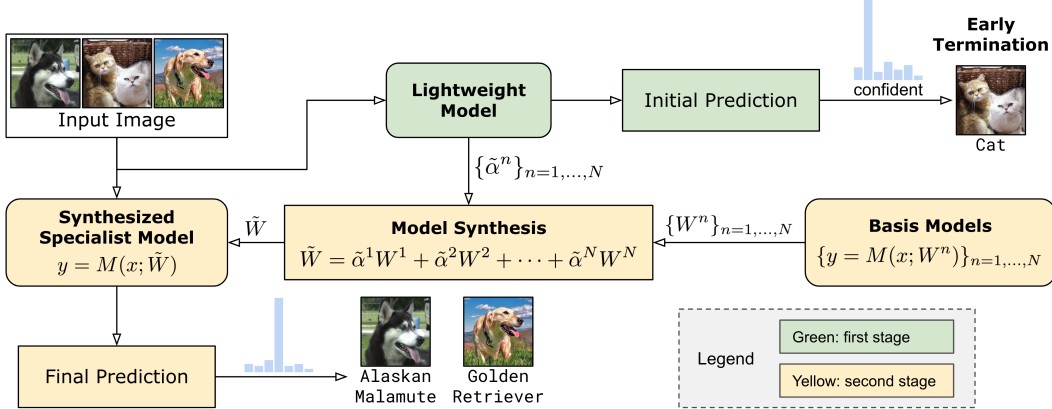

Figure 1: An overview of the BasisNet and more details can be found in Sec. 3.2. For easy images (e.g., distinguishing cat from dogs), lightweight model can give sufficiently accurate predictions thus the second stage could be bypassed. For more difficult images (e.g., distinguishing different breeds of dogs), a specialist model is synthesized following guidance from lightweight model, which is good at recognizing subtle differences to make more accurate predictions about the given images.

Table 1: Comparison with selected efficient networks on ImageNet. Statistics on referenced baselines are cited from original papers. See Appendix C for detailed comparison incl. training recipes.

|  | MAdds (FLOPs) | Top-1 Acc./% |
|---|---|---|
| MobileNetV2 1.0x (Sandler et al., 2018) | 300M | 72.0 |
| MobileNetV3-Large (Howard et al., 2019) | 219M | 75.2 |
| ShuffleNetV2 1.5x (Ma et al., 2018) | 299M | 72.6 |
| ProxylessNas (Cai et al., 2018) | 320M | 74.6 |
| MnasNet-A1 (Tan et al., 2019) | 312M | 75.2 |
| EfficientNet-B2 (Tan & Le, 2019) | 1.0B | 80.1 |
| EfficientNet-B1 (Noisy Student) (Xie et al., 2019)♦♥ | 700M | 80.2 |
| FBNetV3-E (Dai et al., 2020)♣ | 752M | 80.4 |
| OFA (Cai et al., 2019)♦ | 595M | 80.0 |
| **BasisNet-MV3 (Ours)♦♠♡** | **290M** | **80.3** |
| **BasisNet-MV3 + Early Termination (Ours)♦♠♡** | **198M (Avg.)** | **80.0** |

and produce both an initial prediction and a group of combination coefficients. In the second stage, the coefficients are used to combine a set of models, which we call *basis models*, into a single one to process the image and generate the final classification result. The second stage could be skipped if the initial prediction is sufficiently confident. The basis models share the same architecture but differ in some weight parameters, while other weights are shared to avoid overfitting and reduce the total model size.

We validated BasisNet with different generations and sizes of MobileNets and observed significant improvements in inference efficiency. In Table 1 we show comparisons[1] with recent efficient networks on ImageNet classification benchmark. Notably, *without* using early termination, our BasisNet with 16 basis models of MobileNetV3-large only requires 290M Multiply-Adds (MAdds) to achieve 80.3% top-1 accuracy, halving the computation cost of previous state-of-the-art (Cai et al., 2019) without sacrificing accuracy. If we enable early termination, the *average* cost can be further reduced to 198M MAdds with the top-1 accuracy remaining 80.0% on ImageNet.[2]

---

[1]Listed models may use different training recipes (e.g., knowledge distillation♦, extra data♥, custom data augmentation♠, and AutoML-based training hyperparameters search♣, etc.)

[2]Avg cost is reduced since easy inputs are only handled by lightweight model; max remains 290M MAdds.

Our main contributions are summarized below:

- We propose a two-stage model synthesis strategy that combines efficient neural nets, conditional computation, and early termination in a simple new form. Our BasisNet achieves state-of-the-art performance of accuracy with respect to computation budget on ImageNet even *without* early termination; if enabling early termination, the average computation cost can be further reduced.
- We propose an accompanying training procedure for the new BasisNet, which is also effective to improve the performance for some other models (e.g., MobileNets).

## 2 RELATED WORK

**Efficient neural networks**   Different approaches for building efficient networks have been studied. Early effort includes knowledge distillation (Hinton et al., 2015), post-training pruning (Han et al., 2015; Hu et al., 2016) and quantization (Courbariaux et al., 2016; Jacob et al., 2018). Later work distinguishes model complexity (size) and run-time latency (speed) and optimizes for them either with human expertise (Howard et al., 2017; Sandler et al., 2018; Ma et al., 2018; Zhang et al., 2018) and/or neural architecture search (Cai et al., 2018; Tan et al., 2019; Howard et al., 2019; Tan & Le, 2019; Cai et al., 2019; Dai et al., 2020). All these approaches aim at producing a *static* model that is generally efficient but agnostic to inputs. On the contrary, our BasisNet is built on top of any efficient network architectures, and is *dynamically* adaptive based on specific inputs. Also we optimize for model inference speed rather than model size.

**Conditional computation**   Several prior work have explored accelerating inference by skipping part of computation graph based on input-dependent signals. For example, Figurnov et al. (2017) propose a ResNet extension that dynamically adjusts the number of executed layers based on image regions. Teja Mullapudi et al. (2018) propose HydraNet which creates multiple parallel branches across the network, and adopts a soft gating module to selectively activate few branches to reduce inference cost. Shazeer et al. (2017) use mixture of experts with a gating network to choose from thousands of candidates. Recently, Yang et al. (2019) propose conditionally parameterized convolution (CondConv), which applies weighted combinations of convolution kernels. This idea is adopted by several later work (Zhang et al., 2019; 2020; Chen et al., 2020), because it has equivalent expressive power as linear mixture of experts, but requires much fewer computations than combining feature maps. However, one common characteristic of these approaches is that their conditioning modules are inserted before each configurable component (e.g., layer or branch), thus they can only rely on *local* information (i.e., outputs from previous layer) to make dynamic adjustments (Chen et al., 2019). Lacking *global knowledge* may be less ideal because shallower layers cannot benefit from semantic knowledge which is only available from deeper layers. This conceptual distinction is suggested by Chen et al. (2019), and some other work also identified similar issues and have attempted to leverage *global knowledge* in dynamic modulation, in order to ultimately improve model performance. For example, in SkipNet (Wang et al., 2018) a gating network is built to conditionally skip certain layers in the backbone, and the authors report that the best performance comes from a RNN-based gating network because it can access feature maps across multiple layers. Chen et al. (2019) introduce GaterNet where a dedicated deep neural network is used to analyze the inputs before generating input-dependent masks for the filters in backbone network. BasisNet use a lightweight but fully-fledged model to process the inputs and produce dynamic combination signals, thus the model synthesis is relying on semantic-aware *global knowledge*. But different from SkipNet and GaterNet, our lightweight model can synthesize new kernels that do not exist beforehand, rather than simply selecting from available candidates. Another distinction is that by separating conditioning model from backbone, our BasisNet is more flexible and easier to adapt to different architectures and hardware constraints.

**Cascading networks and early exiting**   Since input samples are naturally of varying difficulty, using a single model to equally process all inputs with a fixed computation budget is wasteful. This observation has been leveraged by prior work, e.g., the famous Viola-Jones face detector (Viola & Jones, 2001) built a cascade of increasingly more complex classifiers to achieve real-time execution. Similar ideas were also used in deep learning, e.g., reducing unnecessary inference computations for easy cases in a cascaded system (Gama & Brazdil, 2000; Venkataramani et al., 2015), attaching multiple classification heads on different layers (Teerapittayanon et al., 2016; Leroux et al., 2017;

Marquez et al., 2018; Huang et al., 2018), or cascading multiple models (Park et al., 2015; Bolukbasi et al., 2017). One common limitation in previous work is that only the exit point adapts to the samples but the underlying models remain static. Instead, our BasisNet dynamically adjusts the convolution kernel weights based on the guidance from lightweight model, thus the synthesized specialist can better handle the more difficult cases.

## 3 APPROACH

In general, our BasisNet has two stages: the first stage lightweight model, and the second stage model synthesis from a set of basis models. Given a specific input, the lightweight model generates two outputs, an initial prediction and a group of basis *combination coefficients*. If the initial prediction is of high confidence, the input is presumably easy and BasisNet could directly return the initial prediction and terminate early. But if the initial prediction is less confident (implying the input is difficult), the *coefficients* will be used to guide the synthesis of a specialist model in the second stage. The synthesized specialist will be used for generating a final prediction.

### 3.1 LIGHTWEIGHT MODEL

The lightweight model is a fully-fledged network handling two tasks: (1) generating initial classification prediction and (2) generating combination coefficients for second stage model synthesis. The first is a standard classification thus we only elaborate on the second below. A more complete description is provided in Appendix A.1. Assuming there are $N$ basis models for the second stage and each has $K$ layers, the lightweight model will predict combination coefficients $\alpha \in \mathcal{R}^{K \times N}$

$$\alpha = \phi(\text{LM}(f(x))) \tag{1}$$

where LM stands for *lightweight model* and $\phi$ represents a non-linear activation function. We use softmax by default because it enforces convexity, which promotes sparsity and can lead to more efficient implementations. $f(x)$ represents a transformation of the input image, and we use $f(x) = x$ or $f(x) = \text{DownSampling}(x)$.

### 3.2 BASIS MODEL SYNTHESIS

Our basis models are a collection of model candidates, which share the same architecture but differ in model parameters. By combining basis models with different weights, a specialist network can be synthesized. Various strategies can be used for building basis models, such as mixture of experts or using multiple parameter-efficient patches (Mudrakarta et al., 2018). We explored a few options and found that the recently proposed CondConv (Yang et al., 2019) best fits our needs for building a low-inference cost but high-capacity model.

Specifically, consider a *regular* deep network with image input $x$. Assume the output of the $k$-th convolutional layer is $O_k(x)$, which could be obtained by

$$O_k(x) = \begin{cases} \phi(W_0 * x), & \text{if } k = 0 \\ \phi(W_k * O_{k-1}(x)), & \text{if } k > 0 \end{cases} \tag{2}$$

where $W_k$ represents the convolution kernel at the $k$-th layer and $*$ represents a convolution operation. For simplicity some operations like batch normalization and squeeze-and-excitation are omitted from the notation. In BasisNet, different inputs will be processed by different, input-dependent kernel $\tilde{W}_k$ at $k$-th layer, which is obtained by linearly combining the kernels from $N$ basis models at $k$-th layer, denoted by $\{W_k^n\}_{n=1,\dots,N}$:

$$\tilde{W}_k = \tilde{\alpha}_k^1 \cdot W_k^1 + \cdots + \tilde{\alpha}_k^N \cdot W_k^N \tag{3}$$

where $\tilde{\alpha}_k^n$ represents the weight for the $k$-th layer of the $n$-th basis. We use $\tilde{W}$ and $\tilde{\alpha}$ to emphasize their dependency on $x$. This design allows us to increase model capacity effectively but retain the same number of convolution operations. Besides, since the number of parameters is much less than number of MAdds in a single basis architecture, the combination only marginally increase the computation cost. Using sparse convex coefficients further reduces the combination overhead.

We generally consider convex coefficients, but also studied two special cases in Appendix D.1:

- $\alpha_k$ is the same for all layers. In this case, the combination is *per-model* instead of *per-layer*.
- $\alpha_k$ as an $N$-dimension vector is one-hot encoded. In this case, synthesis becomes model selection.

**Key difference from CondConv**    Our model synthesis mechanism is inspired by CondConv (Yang et al., 2019) but there exists many distinctions. In CondConv the combination coefficients for $k$-th layer are computed following

$$\alpha_k = \phi(\text{FullyConnected}(\text{GlobalAveragePooling}(O_{k-1}(x)))) \tag{4}$$

which means the dynamic kernel can only be synthesized *layer by layer*, because the combination coefficients for next layer depend on output of previous layer. This complicates scheduling of computation thus is not hardware friendly (Zhang et al., 2019). In BasisNet, the coefficients are obtained from the lightweight model, therefore the entire specialist model can be synthesized *all at once*. This enables BasisNet to be easily deployed *to* (or even *across*) different hardware accelerators on edge devices. Besides, BasisNet naturally supports early termination which is infeasible for Cond-Conv. We measured the latency on real device and show that applying model synthesis in a separate stage is more efficient than the layer-by-layer synthesis strategy (See Sec 4.7). Lastly, BasisNet is complementary to CondConv, as we find (in Sec. 4.5) that combining CondConv and BasisNet can further boost prediction accuracy.

## 3.3    TRAINING BASISNET PROPERLY

BasisNet significantly increases model capacity, but the risk of overfitting also increases. We found the standard training procedures used to train MobileNets lead to overfitting on BasisNet. Here we describe a few regularization techniques that are crucial for training BasisNet successfully.

- **Basis model dropout (BMD)** Inspired by Gastaldi (2017), we experimented with randomly shutting down certain basis model candidates during training. It is equivalent to applying DropConnect (Wan et al., 2013) on the predicted coefficient matrix from the lightweight model. We found this approach is extremely effective against "experts degeneration" (Eigen et al., 2013; Shazeer et al., 2017) where the controlling model always picks the same few candidates ("experts").
- **AutoAugment (AA)** AutoAugment (Cubuk et al., 2019) is a search-based procedure for finding specific data augmentation policy towards a target dataset. We find that replacing the original data augmentation in MobileNets (Sandler et al., 2018; Howard et al., 2019) with the ImageNet policy in AutoAugment can significantly improve the model generalizability.
- **Knowledge distillation** Hinton et al. (2015) showed that using soft targets from a well-trained teacher network can effectively prevent a student model from overfitting. We experimented using EfficientNet-B2 with noisy student training (Xie et al., 2019) as teacher to train our BasisNet.

In addition to stronger regularization, we applied a few other tricks in order to properly train BasisNet. Since the lightweight model directly controls how the specialist model is synthesized, any slight changes in the combination coefficients will propagate to the parameter of the synthesized model and finally affect the final prediction. Since we train the two stages from scratch, this is especially troublesome at the early phase when the lightweight model is still ill-trained. To deal with the unstable training, we introduced $\epsilon \in [0, 1]$ to balance between a uniform combination and a predicted combination coefficients from the lightweight model,

$$\alpha' = \epsilon \cdot \frac{1}{N} \cdot \mathbf{1}^{K \times N} + (1 - \epsilon) \cdot \alpha \tag{5}$$

When $\epsilon = 1$ all bases are combined equally while when $\epsilon = 0$ the synthesis is following the combination coefficients. In practice $\epsilon$ linearly decays from 1 to 0 in the early phase of training then remains at 0, thus the lightweight model can gradually take over the control of model synthesis. This approach effectively stabilizes training and accelerates convergence. A concurrent work (Chen et al., 2020) proposed a different strategy (temperature-controlled softmax) to achieve similar goal.

All models in both stages are trained together in an end-to-end manner via back-propagation. In other words, all basis models are trained from scratch by gradients from the synthesized model. The total loss includes two cross-entropy losses for the synthesized model and the lightweight model, respectively, and L2 regularization,

$$L = -\log P(y|x; \tilde{W}) + \lambda(-\log P(y|f(x); W_{\text{LM}})) + \Omega(\{W^n\}_{n=1,\dots,N}, W_{\text{LM}}) \tag{6}$$

where $\lambda$ is the weight for cross-entropy loss from lightweight model ($\lambda = 1$ in our experiments), and $\Omega(\cdot)$ is L2 regularization loss applied to all model parameters. The lightweight model receives gradients from all terms, while basis models are only updated by the first term and regularization.

# 4    EXPERIMENTS

## 4.1    DATASET AND MODEL ARCHITECTURE SETUP

We evaluate BasisNet on the ImageNet ILSVRC 2012 classification dataset (Russakovsky et al., 2015) consisting of 1.28M images for training and 50K for validation. We demonstrate the effectiveness of model synthesis on both MobileNetV2 and MobileNetV3 architectures. In Appendix A and B we give details about the network architectures and training hyperparameters.

For fair comparison, we retrained all models including BasisNet and baselines under the same conditions, and reported the performance with early termination *disabled* except for Sec.4.6. Note that the lightweight model introduces computation overhead for BasisNet, but unless stated otherwise, our reported MAdds statistics for BasisNet always *include* the lightweight model overhead.

## 4.2    COMPARISON WITH MOBILENETS

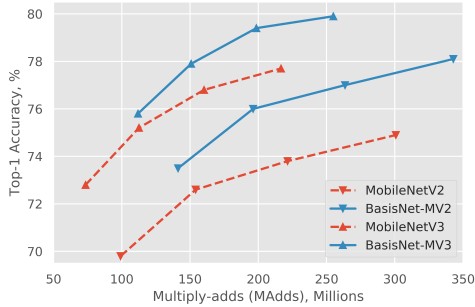
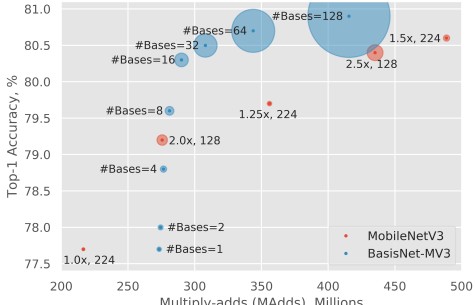

Figure 2: Accuracy-MAdds trade-off comparison of the proposed BasisNet and MobileNet on ImageNet validation set.

Figure 3: Prediction accuracy monotonically increases when more bases are added to the basis models. Details in Appendix D.3.

For both BasisNet-MV2 and BasisNet-MV3, we compute the accuracy-MAdds curves by varying the input image resolution to the synthesized model from {128, 160, 192, 224}. We compute the curves for the MobileNets by varying image resolutions in the same way. As shown in Fig. 2, even with the computation overhead of the lightweight model, our BasisNets consistently outperform the MobileNets with large margins.

## 4.3    THE EFFECT OF REGULARIZATION FOR PROPER TRAINING

In Fig. 4 we show the performance improvements when different regularizations (basis model dropout, AutoAugment, and distillation with EfficientNet-b2 as teacher) discussed in Sec. 3.3 are individually applied to BasisNet-MV2 training, as well as combined altogether. Each regularization helps generalization, and the most effective single regularization is the knowledge distillation. By combining all strategies the validation accuracy increases the most. In Fig. 5 we show the performance curve for BasisNet-MV2 and MobileNetV2 under different regularizations with varied image resolutions. We observe the proposed training procedures also boost performance for the original MobileNet. However, applying the regularization is more effective for BasisNet training, as the performance of BasisNet-MV2 (1.0x224) increases from $74.7\%$ to $78.1\%$ (+3.4).

## 4.4    NUMBER OF BASES IN BASIS MODELS

We varied the number of bases to understand their effect on the model size, inference cost and classification accuracy. Intuitively, the more bases, the more diverse domains the final synthesized model

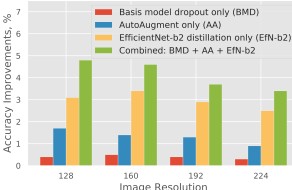 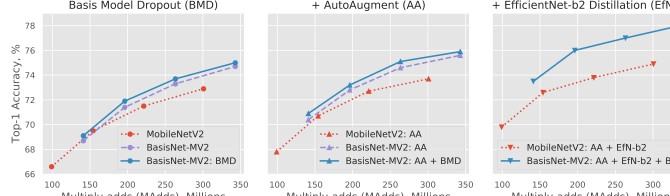

Figure 4: Performance boost with various regularizations on BasisNet-MV2. Combining them altogether gives the largest improvement.

Figure 5: MobileNet and BasisNet training using different regularizations. BasisNet uses MV2-0.5x as its lightweight model and 8 MV2-1.0x for basis models. Input image resolutions vary from {128, 160, 192, 224}. Note that basis model dropout (BMD) is not applicable to MobileNet because it has only one model.

can adapt to. We chose a fix-sized MV3-small (1.0x224) as our lightweight model, and use different numbers of MV3-large (1.0x224) for basis. As shown in Fig. 3, the top-1 accuracy improves monotonically with increased number of bases. With 16 bases, our BasisNet-MV3 achieved 80.3% accuracy with 290M MAdds. The shaded area represents the relative model size (#Params). Note that we explicitly trained a regular MobileNetV3-large with large multiplier and low image resolution (2.5x128), so it has similar model size with BasisNet. We show that BasisNet requires only 2/3 of computations to achieve the same accuracy with the MobileNetV3 counterpart.

## 4.5 COMPARISON WITH CONDCONV

We re-implemented CondConv[3] to directly compare with our BasisNet. We choose MobileNetV3 as backbone, and selected $N = 16$ for both BasisNet and CondConv from layers 11 to 15. We chose MV3-small(1.0x224) as the lightweight model, and disabled early termination for BasisNet. Both models are trained using the same condition as described in Sec. 3.3. It is worth noting that even though the overall computation for BasisNet is larger due to overhead by lightweight model, the synthesized specialist (which is directly responsible for image classification) consumes roughly the same amount of computations for these two models.

The top-1 accuracy for CondConv-MV3 and BasisNet-MV3 is 79.9% and 80.3% respectively, showing the advantage of BasisNet over CondConv. More importantly, we find that BasisNet is more flexible than CondConv. CondConv reports that simultaneously activating multiple routes is necessary for any single input, therefore sigmoid activation has to be used. For BasisNet, we find both sigmoid and softmax work fine (80.0% and 80.3% accuracy respectively), and the latter can lead to sparse and even one-hot combination coefficients (see Sec. 4.8). We also experimented to combine CondConv with BasisNet, and the accuracy is further boosted to 80.5%. However this will prevent model being synthesized all at once (Sec. 3.2) thus diminishing the purpose of developing BasisNet.

## 4.6 REDUCE AVERAGE INFERENCE COST VIA EARLY EXITING

The two stage design of BasisNet naturally supports early termination, since the lightweight model can choose to skip the second stage and returns its prediction directly if confident. We verified the effectiveness on ImageNet validation set with a well-trained BasisNet-MV3 (1.0x224,16 basis) model. We chose the maximum value of softmax probability (Geifman & El-Yaniv, 2017; Huang et al., 2018) as the criterion to measure initial prediction confidence. We split the 50K validation images into multiple buckets according to the sorted top-1 probability, and in Fig. 6 (left) we show the accuracy within each bucket by the lightweight model, synthesized specialist model and a reference MobileNetV3. We observe that for at least one third of images where lightweight model has high prediction confidence, the accuracy gaps between these three models are negligible (< 1%). The BasisNet has clear advantage over MobileNet in all buckets, especially for more difficult (low confidence) cases. We also run a simulation by altering thresholds of prediction confidence: for all images that show confidence higher than a threshold the second stage will be skipped, while for the

---

[3]Our re-implementation of CondConv-MV2 achieved 76.2% accuracy, better than the reported accuracy of 74.6% from Yang et al. (2019). More details in Appendix D.4.

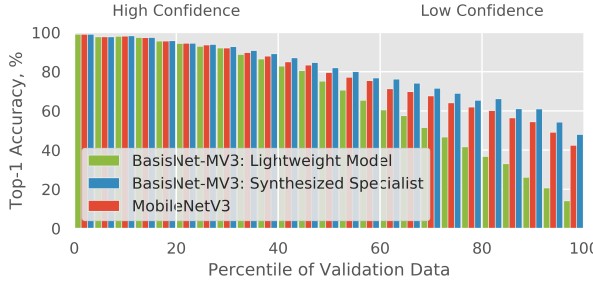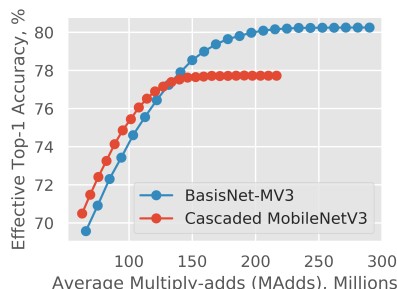

Figure 6: Early exiting can further reduce computation cost without sacrificing accuracy. (Left) Prediction accuracy is comparable for more confident predictions (e.g. top 40%), and the synthesized specialist consistently outperforms regular MobileNet in all buckets; (Right) Simulation of BasisNet-MV3 with early exiting under varying threshold.

Table 2: Latency measurements on Google Pixel 3XL for different models.

| Methods | Accuracy(%) | MAdds(M) | Latency(ms) |
|---|---|---|---|
| BasisNet-MV3 8-routes | 79.6 | 281 | 60.6 |
| BasisNet-MV3 16-routes | 80.3 | 290 | 62.9 |
| – With early termination | 80.0 | 198 (avg.) | 43.6 (avg.) |
| MobileNetV3 (1.25x224) | 79.7 | 356 | 66.3 |
| MobileNetV3 (1.5x224) | 80.6 | 489 | 86.2 |
| CondConv-MobileNetV3 | 79.9 | 253 | 53.1 |

rest the specialist will be synthesized and used. With different thresholds the BasisNet has different cost and accuracy. For fair comparison, we cascade two well-trained MobileNets of the same size as the lightweight model and basis model respectively. Fig. 6 (right) shows that BasisNet achieves better results for the same cost, except when the computation budget is very limited.[4] Particularly for BasisNet, with a threshold of 0.7, 39.3% of all images will skip the second stage thus the average computation cost reduces to 198M MAdds while the overall accuracy remains 80.0%.

### 4.7 ON-DEVICE LATENCY MEASUREMENTS

To validate the practical applicability, we measured the latency of the proposed BasisNet and other baselines on physical mobile device. We choose Google Pixel 3XL and run floating-point models on the big core of the phone's CPU. In Table 2 we show that BasisNet can run efficiently on existing mobile device. Our efficiency conclusion drawn from MAdds also applies to real latency. Specifically, MobileNetV3 with 1.25x and 1.5x multipliers have similar accuracy as BasisNet-MV3 with 8 and 16 routes, while the BasisNet has lower latency. Our two-stage model synthesis design enables early termination, which offers even more efficient execution (as shown by the estimated average latency of 43.6ms).[5] We also measured the latency for CondConv in the same table. CondConv has lower latency than BasisNet primarily because it does not use the first-stage lightweight model. However, the first stage computes basis weights with better results (80.3% vs 79.9%), and gets early termination for free. To more fairly compare with CondConv, one can add a first stage to CondConv but that may lead to (1) extra computation; (2) the overall accuracy cannot go beyond 79.9% which is the accuracy of CondConv. Lastly, BasisNet synthesizes a specialist that has the same architecture as existing state-of-the-art mobile networks. So it has the potential to be supported by any accelerators that are optimized for existing mobile network architectures. We leave exploring this direction as future work.

---

[4]The accuracy of lightweight model prediction is slightly worse than the corresponding MobileNet, shown as the data points on the left end of Fig. 6 (right), because the lightweight model needs to handle two tasks.

[5]With threshold of 0.7 on ImageNet, 39.3% of images can skip second stage thus the estimated average latency is reduced to $0.393 \times 13.7\text{ms} + (1 - 0.393) \times 62.9\text{ms} = 43.6\text{ms}$.

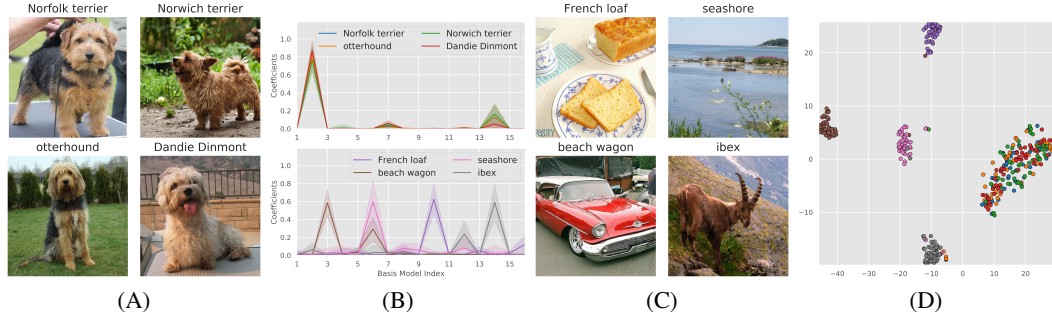

Figure 7: (A,C) Sample images from visually similar or distinct categories. (B) Mean coefficient weights at 15-th layer for selected categories. (D) t-SNE visualization of combination coefficients.

Table 3: Different disturbance applied to the combination coefficients.

| Disturbance | CORRECT | TOP-1 | MEAN | UNIFORM | SHUFFLED |
|---|---|---|---|---|---|
| BasisNet-MV2 | 78.2 | 73.9 (-4.3) | 67.2 (-11.0) | 67.2 (-11.0) | 56.5 (-21.7) |
| BasisNet-MV3 | 79.8 | 77.8 (-2.0) | 69.5 (-10.3) | 69.7 (-10.1) | 58.1 (-21.7) |

### 4.8 UNDERSTANDING THE LEARNED BASISNET MODELS

**Visualizing the specialization of basis models.** We visualized the combination coefficient vectors on ImageNet validation set to better understand the effectiveness of model synthesis. In Fig. 7 we show visually similar and distinct categories, as well as the combination coefficients of 15-th layer. The lightweight model chooses the same specialist to better handle the subtleties between dog breeds, but for visually distinct categories the synthesized models are very different (curves in (B) bottom do not coincide). In Fig. 7 (D) we show the coefficients for all images using t-SNE. The dog categories form a single cluster while the others reside in very different clusters. We also find different bases are activated by fine-grained visual patterns, e.g. fluffy dogs mainly activate 2nd base and short-haired dogs use 14th base. More qualitative examples are shown in Appendix E.

**The importance of optimal basis model synthesis.** To verify the importance of model synthesis, we add disturbances to the predicted combination coefficients. The specialist should be most effective for the corresponding image, and a disturbed synthesis signal would hurt performance. We train BasisNet-MV2 (Accuracy 78.2%) and BasisNet-MV3 (Accuracy 79.8%), and share only the first 7 layers in the basis, then disturb the coefficients $\alpha$ as follows: (1) preserving the highest probable basis model only (TOP-1), (2) uniformly combining all basis models (UNIFORM), (3) using mean weights over entire validation set (MEAN), or (4) randomly shuffling the coefficients within each layer (SHUFFLED). As shown in Table 3, all disturbances lead to inferior performance validating that basis models have varied expertise. SHUFFLED leads to a totally mismatched specialist thus performance drops over 20%. Choosing TOP-1 has the smallest accuracy drop, showing potential for learning model selection which we leave for future work. We also applied the disturbance to individual layers and observed some interesting patterns, as detailed in Appendix D.6.

## 5 CONCLUSION

We present BasisNet, which combines the recent advancements in multiple perspectives such as efficient model design and dynamic inference. With a standalone lightweight model, the unnecessary computation on easy examples can be saved and the information extracted by the lightweight model help synthesizing a specialist network for better prediction. With extensive experiments on ImageNet we show the proposed BasisNet is particularly effective on efficient inference, and BasisNet-MV3 achieves 80.3% top-1 accuracy with only 290M MAdds even without early termination.

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

APPENDICES

## A    DETAILED MODEL ARCHITECTURE OF BASISNET

In this section we describe in details about the proposed BasisNet, including the lightweight model and basis models.

### A.1    LIGHTWEIGHT MODEL

For BasisNet-MV2, the lightweight model follows the architecture described in Table 2 of Sandler et al. (2018), and we use multiplier of $0.5$ and input image resolution of $128$. The lightweight model has a computation overhead of 30.3M MAdds and a model size of 1.2M parameters.

For BasisNet-MV3, we use MobileNet v3-small for our lightweight model as described in Table 2 of Howard et al. (2019), and we use multiplier of $1.0$ and input resolution of $128$ or $224$ for different experiments. The model size for the lightweight model is 2.5M parameters regardless of input image resolutions. With $128 \times 128$ image, the lightweight model has 19.9M MAdds computation overhead, and with $224 \times 224$ image the computation overhead is 56.5M MAdds.

As described in Sec.3.1, the lightweight model has two tasks, one for initial classification prediction and the other for combination coefficients prediction. The first task is similar with any regular classification task, and can be formally described as:

$$\hat{y} = \text{LM}(f(x); W_{\text{LM}}) \tag{7}$$

Note that the two tasks share all but the final classification layer, thus the extra computation for predicting the combination coefficients is negligible.

### A.2    LAYER NUMBERING IN BASIS MODELS

For BasisNet-MV2, the basis models follow the architecture described in Table 2 of Sandler et al. (2018). For simplicity in notation, we sequentially number all the layers starting from L0 until the final $k$-way classification layer as L20, e.g. the first `conv2d` layer in Table 2 is L0 and the `avgpool,7x7` layer is L19.

For BasisNet-MV3, the basis models follow the MobileNet v3-large architecture described in Table 1 of Howard et al. (2019). We also sequentially number all the layers starting from L0 until the final $k$-way classification layer as L19, e.g. the first `conv2d` layer is L0 and the `pool,7x7` layer is L17.

### A.3    DETAILED ARCHITECTURES FOR DIFFERENT EXPERIMENTS

Here we describe in detail the models that used in different experiments. Unless stated otherwise, we use the following settings as default for BasisNet-MV2 and BasisNet-MV3:

- For MobileNetV2 experiments, the first-stage lightweight model is MobileNetV2 with 0.5x multiplier and input image resolution of 128 (MV2, 0.5x128) and the second stage has 8 basis models of MobileNetV2 1.0x with image resolution of 224 (MV2, 1.0x224). Basis models share parameters in layers from 1 to 10 and final classification layer, and differ in parameters in layers 11 to 17.
- For MobileNetV3, the lightweight model is MobileNetV3-small with 1.0x multiplier and input image resolution of 128 (MV3-small, 1.0x128). The second stage has 16 basis models of MobileNetV3-large with 1.0x multiplier and resolution of 224 (MV3-large, 1.0x224), and they share parameters in first 7 and last 2 layers, and differ in parameters in layers 8 to 15.

**Comparison with MobileNets (Sec. 4.2)**    We use BasisNet-MV2 with 8 bases and the lightweight model is MV2 (0.5x128). Each basis model is a MV2 (1.0x224) and they only differ in parameters from L11-17. The basis models dropout rate is $1/8$.

For BasisNet-MV3, we use 16 bases each of a MV3-large (1.0x224), and the lightweight model is MV3-small (1.0x128). All basis models share parameters except for in layers L8-15. The basis model dropout rate is $1/16$.

**The effect of regularization for proper training (Sec. 4.3)**    We use the same model architectures for BasisNet-MV2 and BasisNet-MV3 with Sec. 4.2.

**Number of bases in basis models (Sec. 4.4)**    We use BasisNet-MV3 with different number of basis models, but each is a MV3-large (1.0x224). The lightweight model is MV3-small (1.0x224) and all basis models share parameters except for layers L11-15. For BasisNet with no more than 8 bases we use basis model dropout rate of $1/8$ and for all others (16 to 128 bases) we use a basis model dropout rate of $1/16$.

**Comparison with CondConv (Sec. 4.5)**    For BasisNet-MV3, we use 16 basis models each of a MV3-large (1.0x224), and the lightweight model is MV3-small (1.0x224). All basis models share parameters except for layers L11-15. The basis model dropout rate is $1/16$.

**Reducing average inference cost via early exiting (Sec. 4.6)**    We use the same BasisNet-MV3 model as in Sec. 4.5.

## B    IMPLEMENTATIONS AND TRAINING HYPERPARAMETERS

Our project is implemented with TensorFlow (Abadi et al., 2016). Following Sandler et al. (2018) and Howard et al. (2019), we train all models using synchronous training setup on 8x8 TPU Pod, and we use standard RMSProp optimizer with both decay and momentum set to 0.9. The initial learning rate is set to 0.006 and linearly warms up within the first 20 epochs. The learning rate decays every 6 epochs for BasisNet-MV2 (4.5 epochs for BasisNet-MV3) by a factor of 0.99. The total batch size is 16384 (i.e. 128 images per chip). For stabilizing the training, as described in Section 3.3 we keep $epsilon = 1$ for the first 10K training steps then linearly decays to 0 in the next 40K steps. We also used gradients clipping with clip norm of 0.1 for BasisNet-MV3. In general, all BasisNet and reference baseline models are trained for 400K steps. We set the L2 weight decay to 1e-5, and used the data augmentation policy for ImageNet from AutoAugment (Cubuk et al., 2019). We choose the checkpoint from (Xie et al., 2019) as our EfficientNet-b2 teacher model for distillation, and for BasisNet-MV3 both lightweight model and all basis models are trained with teacher supervision. For BasisNet-MV2, we only distill the basis models but use the groundtruths labels without label smoothing for training the lightweight model. For basis models dropout, we use dropout rate of $1/8$ for all BasisNets with no more than 8 bases, and use $1/16$ for the rest which has 16 or more bases. Following Howard et al. (2019), we also use exponential moving average with decay 0.9999 and set the dropout keep probability to 0.8.

## C    COMPARISON WITH OTHER EFFICIENT NETWORKS

In Table 4 we show a more complete comparison with recent efficient neural networks on ImageNet classification benchmark. For baselines we directly use the statistics from the corresponding original papers, even though the training procedures could be very different. Some common tricks in literature include knowledge distillation♦, training with extra data♥, applying custom data augmentation♠, or using AutoML-based learned training recipes (hyperparameters)♣. Different models may choose subsets of these tricks in their training procedure. For example, Xie et al. (2019) use 3.5B weakly labeled images as extra data and use knowledge distillation to iteratively train better student models. CondConv (Yang et al., 2019) use AutoAugment (Cubuk et al., 2019) and mixup (Zhang et al., 2017) as custom data augmentation. Wei et al. (2020) reported in a concurrent work that combining AutoAugment and knowledge distillation can have even stronger performance boost, because soft-labels from knowledge distillation helps alleviating label misalignment during aggressive data augmentation. In FBNetV3 (Dai et al., 2020) the training hyperparameters are treated as components in the search space and are obtained from AutoML-based joint architecture-recipe search. OFA (Cai et al., 2019) use the largest model as teacher to perform knowledge distillation to improve the smaller models. Notably, in our main paper, unless stated otherwise, we always reported the statistics from our re-implementations, thus the comparison in our ablation studies are fair, but some results might be inconsistent with this table. It is also worth mentioning that even though we did not explicitly use extra data for training BasisNet, the teacher model check-

Table 4: Complete comparison of different efficient networks on ImageNet classification. For baselines, we cite statistics on ImageNet from original papers. Our results are bolded.

| | MAdds (FLOPs) | Top-1 Acc./% |
|---|---|---|
| MobileNetV2 1.0x (Sandler et al., 2018) | 300M | 72.0 |
| CondConv-MobileNetV2 1.0x (Yang et al., 2019)♠ | 329M | 74.6 |
| DY-MobileNetV2 1.0x (Chen et al., 2020)♠ | 313M | 75.2 |
| MobileNetV3-Large (Howard et al., 2019) | 219M | 75.2 |
| Dy-MobileNetV3-Large (Zhang et al., 2020) | 228M | 77.1 |
| ShuffleNetV2 1.5x (Ma et al., 2018) | 299M | 72.6 |
| EfficientNet-B0 (Tan & Le, 2019) | 390M | 77.1 |
| EfficientNet-B0 (Noisy Student) (Xie et al., 2019)♦♥ | 390M | 78.1 |
| EfficientNet-B0 (AA + KD) (Wei et al., 2020)♦♠ | 390M | 78.0 |
| CondConv-EfficientNet-B0 (Yang et al., 2019)♠ | 413M | 78.3 |
| ProxylessNas (Cai et al., 2018) | 320M | 74.6 |
| FBNetV2-L1 (Wan et al., 2020) | 325M | 77.2 |
| FBNetV3-A (Dai et al., 2020)♣ | 343M | 78.0 |
| MnasNet-A1 (Tan et al., 2019) | 312M | 75.2 |
| CondConv-MnasNet-A1 (Yang et al., 2019)♠ | 325M | 76.2 |
| EfficientNet-B2 (Tan & Le, 2019) | 1.0B | 80.1 |
| EfficientNet-B1 (Noisy Student) (Xie et al., 2019)♦♥ | 700M | 80.2 |
| FBNetV3-E (Dai et al., 2020)♣ | 752M | 80.4 |
| OFA (Cai et al., 2019)♦ | 595M | 80.0 |
| **BasisNet-MV3 (Ours)♦♠♡** | **290M** | **80.3** |
| **BasisNet-MV3 + Early Termination (Ours)♦♠♡** | **198M (Avg.)** | **80.0** |

♥: Training with extra data
♦: Knowledge distillation
♣: AutoML-based training hyperparameters
♠: Custom data augmentation

point that we used for knowledge distillation is from noisy student training (Xie et al., 2019), thus our model may indirectly benefit from the extra data.

# D   MORE QUANTITATIVE EXPERIMENTS

## D.1   CONVEX COMBINATION: SPECIAL CASES

**Per-model model synthesis**   We experiment with BasisNet-MV3 for per-model synthesis and per-layer synthesis. Specifically, when lightweight model predicts a single vector of combination coefficients for all layers, i.e. $\alpha_1 = \alpha_2 = \cdots = \alpha_K \in \mathcal{R}^N$, it can be seen as a per-model synthesis.

Note that per-model synthesis of BasisNet is still different from HydraNets (Teja Mullapudi et al., 2018), as the branches in HydraNets span across multiple layers and do not fuse in the middle; instead, in BasisNet the convolution kernels are obtained from linear combination of basis models for each layer.

We use BasisNet-MV3 with 8 bases and a lightweight model of MV3-small (1.0x224), and share all layers in basis models except for L11-15. Interestingly both per-model BasisNet and per-layer BasisNet have the same performance, 79.6% top-1 accuracy on ImageNet validation set, implying the combination coefficients across layers may have high correlations for BasisNet-MV3.

However, we also experiment with per-model variation of BasisNet-MV2 with 8 bases and using a lightweight model of MV2 (0.5x128), and share all layers in basis models except for L11-17. It turns out training per-model BasisNet-MV2 is more challenging as the model always collapses after roughly 30K steps in our multiple attempts. We suspect that training per-model model synthesis is generally more difficult as it has stronger constraints on the basis models, and it may depend on the

base architectures (MobileNetV2 or MobileNetV3). We leave further analysis as future work, and recommend per-layer combination as the default choice.

**Model selection instead of model synthesis**   When the predicted combination coefficients are one-hot encoded, the model synthesis can be simplified as model selection as only one basis model will be selected for a particular layer. We experimented with BasisNet-MV3 with 8 bases, and the lightweight model is MV3-small (1.0x128). Basis models share all layers except for L8-15, and the original BasisNet-MV3 has an accuracy of 79.8% under this setting. After training for 100K steps we froze the lightweight model and transformed the predicted combination coefficients into one-hot embedding, then continued training the basis models. The resulting BasisNet finally achieved 78.5% accuracy. This is +0.7% better than post-processing a well-trained BasisNet (77.8%) implying the potential for training model selection end-to-end.

We leave more careful finetuning for the model selection as future work, but emphasize that model selection has potential to further reduce latency in practice from a model loading I/O perspective.

## D.2   Detailed comparison with MobileNets (Sec. 4.2)

Table 5: Detailed comparison of BasisNet-MV2 with MobileNetV2.

| Model | Preprocess | Distillation | # Bases (BMD) | 128 | 160 | 192 | 224 |
|---|---|---|---|---|---|---|---|
| MobileNetV2 | regular | None | N/A | 66.6 | 69.5 | 71.5 | 72.9 |
| MobileNetV2 | AA | None | N/A | 67.8 | 70.7 | 72.7 | 73.7 |
| MobileNetV2 | AA | MV2 1.4x | N/A | 68.8 | 71.4 | 72.4 | 73.1 |
| MobileNetV2 | AA | EfN-b2 | N/A | 69.8 | 72.6 | 73.8 | 74.9 |
| BasisNet-MV2 | regular | None | 8 (0) | 68.6 | 71.4 | 73.3 | 74.7 |
| BasisNet-MV2 | AA | None | 8 (0) | 70.4 | 72.8 | 74.6 | 75.6 |
| BasisNet-MV2 | regular | EfN-b2 | 8 (0) | 71.8 | 74.8 | 76.2 | 77.2 |
| BasisNet-MV2 | regular | None | 8 (1/8) | 69.1 | 71.9 | 73.7 | 75.0 |
| BasisNet-MV2 | AA | None | 8 (1/8) | 70.9 | 73.2 | 75.1 | 75.9 |
| BasisNet-MV2 | AA | MV2 1.4x | 8 (1/8) | 72.3 | 73.8 | 74.7 | 75.4 |
| BasisNet-MV2 | AA | EfN-b2 | 8 (1/8) | 73.5 | 75.9 | 77.0 | 78.1 |

Here we show original data of Fig. 4 so readers can get the exact accuracy numbers more easily. Specifically we show the model performance with different regularizations at 4 different image resolutions {128, 160, 192, 224} in the last four columns. We compare the data augmentation (Preprocess, *regular* represents the Inception preprocess as in Sandler et al. (2018); Howard et al. (2019), and *AA* represents AutoAugment from Cubuk et al. (2019)), distillation with different teachers (MV2 1.4x represents MobileNetV2 with 1.4x multiplier, EfN-b2 represents EfficientNet-b2 model from Xie et al. (2019)), and basis model dropout.

We experimented with different teacher network to distill the BasisNet. Note that the MobileNetV2 1.4x teacher we used is from Sandler et al. (2018) and has accuracy of 74.9%, and our BasisNet achieves even higher accuracy of 75.4% than the teacher. We also experimented different variations of EfficientNet (b0, b2, b4, b7) and find that models trained with EfficientNet-b2 has the best performance, and using even better teacher network does not bring performance gain to the BasisNet. We suspect this is related to the gap between teacher and student network as reported in Mirzadeh et al. (2020).

## D.3   Detailed experiments for number of bases in basis models (Sec. 4.4)

Here we presents the original data for Fig. 3, so readers can get the exact accuracy numbers more easily. Notably, we find that BasisNet-MV3 with 16 bases is a good balance between model accuracy and computation budget, achieving 80.3% top-1 accuracy with $290M$ Madds. This table also shows that BasisNet technique optimizes MAdds at the expense of model size.

Table 6: Detailed comparison of BasisNets with different number of bases.

| Model | #MAdds/M | #Params/M | Accuracy/% |
|---|---|---|---|
| MV3 (1.0x224) | 217 | 5.45 | 77.7 |
| MV3 (1.25x224) | 356 | 8.22 | 79.7 |
| MV3 (1.5x224) | 489 | 11.3 | 80.6 |
| MV3 (2.0x128) | 276 | 19.1 | 79.2 |
| MV3 (2.5x128) | 435 | 29.0 | 80.4 |
| #Bases=1 | 273 | 8.07 | 77.7 |
| #Bases=2 | 274 | 9.19 | 78.0 |
| #Bases=4 | 277 | 11.4 | 78.8 |
| #Bases=8 | 281 | 15.9 | 79.6 |
| **#Bases=16** | **290** | **24.9** | **80.3** |
| #Bases=32 | 308 | 42.8 | 80.5 |
| #Bases=64 | 344 | 78.6 | 80.7 |
| #Bases=128 | 416 | 150.3 | 80.9 |

## D.4 DETAILED COMPARISON WITH CONDCONV (SEC. 4.5)

In Sec. 4.5 we show a comparison of the proposed BasisNet with CondConv. Specifically, we implement the CondConv routing function as described in Yang et al. (2019) for MobileNetV2 and MobileNetV3. We first compare our re-implementation of CondConv-MobileNetV2 with the original paper to validate the correctness of our implementation. Note that with slightly different hyperparameter choices (for example, we enabled exponential moving average, we used AutoAugment only but not mixup (Zhang et al., 2017) for data augmentation, and our model is trained 8x8 TPU, etc.), our re-implementation achieves better accuracy than reported in the original paper.

To compare with our BasisNet, we select a BasisNet-MV3 with 16 bases and run experiments with a CondConv-MobileNetV3. As shown in Table 7, the CondConv-MobileNetV3 has 17.9M MAdds computation overhead and with sigmoid activation the overall accuracy is 79.9%. For our BasisNet-MV3, the lightweight model is MV3-small (1.0x224) which has 55.6M computation overhead, and with softmax activation the final model achieves 80.3% accuracy. We also changed the activation function for our BasisNet-MV3 system to sigmoid, but find that the accuracy drops slightly to 80.0%. For the experiments that combines CondConv and lightweight model, we calculate the summation of the predicted pre-activation logits from both CondConv routing function and the lightweight model, then apply softmax activation to the sum to get the coefficients for synthesizing basis models. This combination strategy has the same computation overhead with using a lightweight model only ($55.6M$), but the final performance increases to 80.5%.

Table 7: Comparison of BasisNet with CondConv.

| Model type | CondConv-MobileNetV3 | BasisNet-MV3 | BasisNet-MV3 | BasisNet-MV3 + CC-MV3 |
|---|---|---|---|---|
| Activation | Sigmoid | Softmax | Sigmoid | Softmax |
| Computation overhead | 17.9 M | 55.6 M | 55.6 M | 55.6 M |
| Top-1 accuracy | 79.9% | 80.3% | 80.0% | 80.5% |

## D.5 MODEL SYNTHESIS WITH VARYING SIZED LIGHTWEIGHT MODEL

We studied the performance of BasisNet with lightweight model of different size. Here the size is measured by the Multiply-adds (MAdds) as we pay more attention to the inference cost. We experimented with a BasisNet-MV3 of MV3-large (1.0x224) with 8 bases. The lightweight model is MV3-small, and we experimented with two hyperparameters, i.e. the input image resolution to lightweight model (`{128, 160, 192, 224}`) and the multiplier (`{0.35, 0.5, 0.75, 1.0}`). As shown in Figure 8, even an extremely efficient lightweight model (MV3-small (0.35x128), computation overhead of 13.8M Madds) can lead to a performance boost from 77.7% to 78.9% (+1.2%).

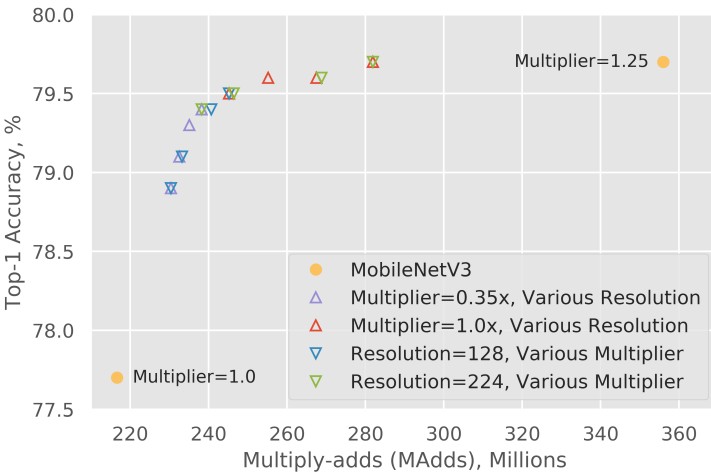

Figure 8: BasisNet-MV3 with lightweight model of different sizes (#MAdds).

This experiment shows that resolution and multiplier can have an equivalent effect as reported in Sandler et al. (2019) and a lightweight model with a smaller computation overhead can bring most of the performance gain. Thus it might be more beneficial to scale the model multiplier and resolution coordinately (Tan & Le, 2019).

### D.6 EFFECT OF MODEL SYNTHESIS AT DIFFERENT LAYERS

Table 8: Performance drop when SHUFFLED disturbance was applied at different layer.

| Disturbed Layer | 8 | 9 | 10 | 11 | 12 | 13 | 14 | 15 | 16 | 17 | 18 | Ref. |
|---|---|---|---|---|---|---|---|---|---|---|---|---|
| BasisNet-MV2 | 78.1 | 78.1 | 78.0 | 78.0 | 77.7 | 77.4 | 77.2 | 76.0 | 76.0 | 76.1 | 77.2 | 78.2 |
| | (-0.1) | (-0.1) | (-0.2) | (-0.2) | (-0.5) | (-0.8) | (-1.0) | (-2.2) | (-2.2) | (-2.1) | (-1.0) | |
| BasisNet-MV3 | 79.6 | 79.6 | 79.6 | 79.4 | 79.0 | 78.3 | 77.9 | 76.4 | 79.1 | – | 76.6 | 79.8 |
| | (-0.2) | (-0.2) | (-0.2) | (-0.4) | (-0.8) | (-1.5) | (-1.9) | (-3.4) | (-0.7) | | (-3.2) | |

We also apply disturbances as described in Sec. 4.8 on each individual layer. As shown in Table 8, we find the layers closer to the final classification layer have more impacts, as the accuracy drop is more significant. Interestingly, the regular convolutional layer right after the *residual bottleneck layers* Sandler et al. (2018); Howard et al. (2019) (e.g. the 18-th layer of MobileNetV2 and the 16-th layer of MobileNetV3) seems less sensitive towards inputs.

## E MORE QUALITATIVE VISUALIZATIONS

### E.1 TOP CATEGORIES HANDLED BY DIFFERENT BASIS MODELS

In Figure 9 we show several most strongly activated categories for four different basis models on ImageNet validation set. Specifically we trained BasisNet-MV3 with 16 bases, and checked the mean weights at the last non-sharing layer (L15) and show the categories that have the highest mean weights. It is clear that the lightweight model captures the fine-grained visual similarity, for example the base 2 seems to handle the fluffy dogs while the base 14 is more about short-haired dogs. Another example is for base 13 that a clear grid pattern can be found in the images, but semantically these categories are loosely related.

### E.2 COMBINATION COEFFICIENTS FOR VISUALLY SIMILAR CATEGORIES

In Figure 10 we show 10 categories regarding different types of cars and the mean predicted combination coefficients for these categories in all layers. Obviously the lightweight model assigns similar

coefficients for various cars, implying the effectiveness of the lightweight model. For example, we see that in Layer 14 almost all cars are relying on base 8, and in L15 all cars use a combination of base 3 and base 6. Quantitatively BasisNet over these 10 categories have an accuracy of 76.6%, but a corresponding regular MobileNetV3 has only 73.2%.

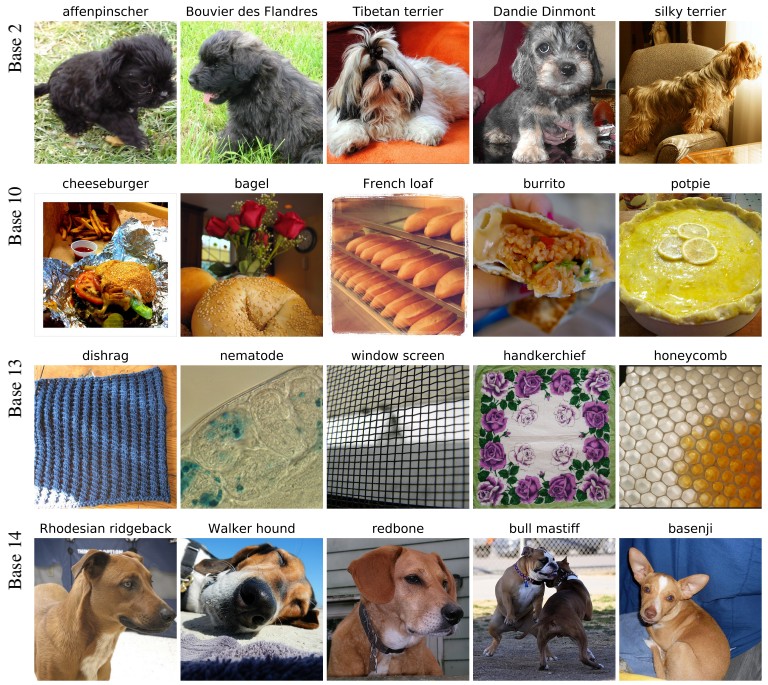

Figure 9: Categories with highest mean coefficients for different basis models.

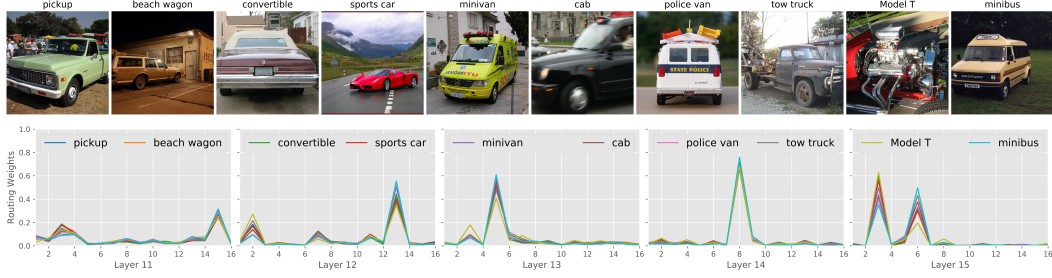

Figure 10: Visualization of predicted combination coefficients for similar categories over all layers.

