# OpenReview forum: "BasisNet: Two-stage Model Synthesis for Efficient Inference"
_ICLR.cc/2021/Conference — Reject_

### Official Review · AnonReviewer3 · 2020-10-27
**A solid paper but maybe with some practical issues**

**Rating:** 6
**Confidence:** 4

**Review:**

This work shows a hierarchical approach of classification by training a set of basis networks and a light-weighted network to combine them linearly for better accuracy given the same MAdds (or vice).  However, there are three problems the reviewer likes to clarify:

(1) What's the optimal batch size at the inference, given that the network is trained at mb=128? If the mb=1, like in a common inference use case, does the basis model require the model synthesis for each image? If the extra MAdds to synthesize a model is ~16M plus the energy cost to load from the device memory, which is actually much larger than MAdd (please see below), the idea seems to be impractical in hardware. On the other end, if mb = 50k when inferencing the entire test dataset, no model synthesizing is needed, but the accuracy of a combination of the BasisNet should not do better than the backbone network as a single inference rendering no extra capacity.  Thus, the authors should add some comments on the accuracy and cost dependences on the inference batch size.

(2) MAdd is a good indicator of hardware efficiency for a model but definitely not the only one from the point of view of a hardware designer. For example, moving one weight from off-chip DRAM or on-chip memory for MAadd could be 10-100 times more expensive than the MAdd (Fig.22, Sze, Vivienne, et al. "Efficient processing of deep neural networks: A tutorial and survey." Proceedings of the IEEE 105.12 (2017): 2295-2329). The cost of synthesizing the model for each batch of data therefore might be expensive unless all 16 weights to be combined can be stored in the nearest RF of each PE. Could the authors show some advantages of using BasisNet on TPU or other hardware optimized for it?

(3) One minor issue on claiming the early termination as an advantage of the BasisNet over the Mobilenets. This does not seem to be fair: if there's no limitation of storing weights on-chip, MB3 could also store a smaller network to inference the easier samples and improve the efficiency in the same way.

Other than these three points, the paper is well written and enough experiments are done to support the authors' point that the BasisNet can provide better accuracy and MAdds than the MobileNets.

---

> ### Author Response · Authors · 2020-11-24
> **Reply to AnonReviewer3**
>
> Thank you for your thorough comments, and we try to answer your questions as below:
> 1. BasisNet will be using batch size of 1 during inference, because the input-dependent model synthesis brings the performance boost as each input will be handled by its own specialist. We performed new experiments on existing physical devices (**BasisNet can run efficiently on existing mobile devices** please see the general feedback on practicality for more details) to show that the real latency follows the similar trends of estimation by MAdds, and our proposed BasisNet is practically efficient and beneficial.
> 2. Regarding the fair comparison between BasisNet and cascaded MobileNet, we show in Figure. 6 (right) of the experiments as suggested. Specifically, the x-axis represents the effective computation budgets, and y-axis represents the estimated accuracy. Therefore, if we look at the same computation cost, the two curves represent fair comparison of BasisNet and cascaded MobileNets.
> Note that except for with very limited budgets (MAdds < 150M), BasisNet consistently outperforms cascaded MobileNet (same computation cost, higher accuracy). We think the reason is that some information is wastefully discarded in cascaded MobileNets settings. In other words, even if the smaller MobileNet cannot confidently make a prediction, the intermediate result that leads to the prediction is still useful. Such a result is leveraged in BasisNet to synthesize a specialist to boost the performance.
>
> We want to thank you again for the time and thoughtful suggestions, and hopefully our explanation and the new experiments could address your concerns.

---

### Official Review · AnonReviewer4 · 2020-10-27
**#Official Review 4**

**Rating:** 3
**Confidence:** 5

**Review:**

This paper present an efficient network, named BasisNet, which combines recent advancements in efficient neural network architectures, conditional computation, and early termination. BasisNet can be applied to any network architectures. BasisNet shows state-of-the-art ImageNet performance in mobile setting.

My main concern is about the novelty of the paper. The authors also claim that they simply combine three kind of works together and achieve a good performance with bag of training tricks (including AA and KD). The main claimed contribution is the basis model synthesis. However, it is just a simple extension of CondConv [1] and DynamicConv [2,3] to the whole model, where the dynamic inference of whole model is also not new [4,5]. The early exit technique is also widely exploited in previous works and the early exit method here is similar to BrachyNet [6]. Thus, the novelty is very limited.

In addition to the novelty, another worry is about the actual inference latency and the practicability. To execute BasisNet for each input image, we first need to run the lightweight model to generate $\alpha$ and initial prediction. Then the weights of N basis model are loaded into the memory and combined using $\alpha$. The synthesized weights are assigned to the BasisNet layer by layer. This process will cost large latency and memory since it must be done online and one by one. In addition, deciding whether eary exit also cost some time. The huge model size further limits practicability of the method.

Overall, this paper is more like a technical report with bag of tricks but very limited novelty. Nothing new is contributed to the community. Since ICLR is a top-tier conference for presenting excellent ideas and works, I think this paper is below the bar of ICLR.

[1] Yang, Brandon, et al. "Condconv: Conditionally parameterized convolutions for efficient inference." Advances in Neural Information Processing Systems. 2019.
[2] Chen, Yinpeng, et al. "Dynamic convolution: Attention over convolution kernels." Proceedings of the IEEE/CVF Conference on Computer Vision and Pattern Recognition. 2020.
[3] Zhang, Yikang, et al. "DyNet: Dynamic Convolution for Accelerating Convolutional Neural Networks." arXiv preprint arXiv:2004.10694 (2020).
[4] Chen, Zhourong, et al. "You look twice: Gaternet for dynamic filter selection in cnns." Proceedings of the IEEE Conference on Computer Vision and Pattern Recognition. 2019.
[5] Cheng, An-Chieh, et al. "InstaNAS: Instance-Aware Neural Architecture Search." AAAI. 2020.
[6] Teerapittayanon, Surat, Bradley McDanel, and Hsiang-Tsung Kung. "Branchynet: Fast inference via early exiting from deep neural networks." ICPR, 2016.

---

> ### Author Response · Authors · 2020-11-24
> **Reply to AnonReviewer4**
>
> Thank you for sharing the relevant papers! We want to emphasize our contributions as below:
> 1. Properly training conditional computation is *not trivial*, as in [1] and [2] they both use similar architectures but only achieve moderate improvements. In Section 3.3, we have extensive experiments to prove that knowledge distillation and data augmentation are critical for proper training, and we achieved **state-of-the-art performance under 300M MAdds budget** (80.3% at 290M MAdds, without early stopping). While individual component  techniques are not new, *before our paper there is no good practice on how to effectively train such high-capacity dynamic neural networks*. Please keep in mind that our training procedure significantly improves CondConv results than what is reported in the original paper, which we report in our paper for fair comparison. We think our discovery is essential for the community to realize the basic combination idea can produce state-of-the-art results, and will be very useful for the community in the future when working with dynamic/conditional neural networks.
> 2. The proposed two-stage model synthesis provides more flexibility for deploying models on edge devices.
>     - It enables early stopping (further improves efficiency without sacrificing performance)
>     - The synthesized specialist has the same network architecture as existing state-of-the-art mobile network architecture. So it has the potential to be supported by any accelerators that are optimized for existing mobile network architectures.
>     - It offers the potential for parallel execution (two stages on separate computation cores if available) for streaming data.
>
>   None of these merits can be achieved by prior work, e.g. CondConv.
> 3. We performed extra experiments to measure the latency on physical devices (**BasisNet can run efficiently on existing mobile devices** please see the general feedback on practicality for more details), and show that the proposed BasisNet is actually feasible and beneficial.
> 4. Lastly, we want to clarify that our BasisNet is not assigning weights in a layer-by-layer fashion. Thanks to the lightweight network, we obtain the synthesizing coefficients for all layers all at once, and can obtain the weights for the specialist model in one step. This is actually one of the advantages of BasisNet over CondConv, because the latter has to synthesize the weights one layer at a time, and that leads to  the difficulty of early stopping.
>
> Again we appreciate your time and comments, and hopefully our explanation and new experiments could address your concerns.
>
> [1] Yang, Brandon, et al. "Condconv: Conditionally parameterized convolutions for efficient inference." Advances in Neural Information Processing Systems. 2019.
> [2] Chen, Yinpeng, et al. "Dynamic convolution: Attention over convolution kernels." Proceedings of the IEEE/CVF Conference on Computer Vision and Pattern Recognition. 2020.

---

### Official Review · AnonReviewer2 · 2020-10-28
**The overall method is easy to understand**

**Rating:** 7
**Confidence:** 5

**Review:**

This paper proposes a framework for acquiring an efficient network that essentially consists of an ensemble of networks.

Strength:
1. The overall method is easy to follow and understand.
2. Good performance with practical appealing properties. Results outperform recent SOTA like the noisy student.

Weakness:
1. Lacking enough analysis of 'global knowledge' or 'global view'. The authors repeatedly highlight that their method has the advantage of capturing 'global knowledge' when compared to related works. However, there is only an intuitive understanding of global knowledge: the knowledge that is semantic-aware and ready for prediction in the lightweight model is global knowledge. The concept is vague, confusing, and lacks a clear definition. What knowledge can be called global or local? Is there a mathematical definition for such a concept? The authors need to carefully address this issue as the concept is a key motivation but not properly described. Besides, the authors need to add experiments to support their claim that global knowledge is indeed better than previous local knowledge in some aspects (sec3.2 Key difference).
2. Lacking simple baseline results or ablation results. Baseline A: Directly uniform sampling from basis networks. That is, always fixing \epsilon to a value close to 1 in Eq. (5). This baseline is to validate the effectiveness of the proposed synthesizing module. Baseline B: Following the cascade mbnet used in Fig. 6 right, the baseline is that applying joint training (Eq6) to cascade mbnet. This baseline is to show the improvements brought by the joint training framework.

---

> ### Author Response · Authors · 2020-11-24
> **Reply to AnonReviewer2**
>
> Thank you for your insightful reviews. Please see our responses below.
> 1. Our intuitive hypothesis about “global knowledge” is mainly inspired from [1], and we follow the similar definitions with them when referring to “global knowledge” and “local knowledge”. The main distinction about global or local, is whether the conditional information is rich enough to accomplish some high-level semantic tasks. For example, when the lightweight model provides signals for synthesizing the main network, the feature it relies on is also capable of making decent predictions on the image category (auxiliary loss). We call such information “global” because intuitively the model needs to process the inputs in its entirety to accomplish the goal. On the contrary, when we refer to “local knowledge” we mean that the feature is restricted and intermediate. For example, in prior work such as CondConv, the “synthesis” is accomplished layer-by-layer, therefore when earlier layers start to produce synthesis signals, they do not have any knowledge about the higher-level representations.
> We have empirical evaluations on these two synthesizing philosophies, and the comparison is shown in supplementary D.4. Specifically, with the same architectures and training protocols, BasisNet outperforms the corresponding CondConv by 0.4% (80.3% vs. 79.9%). We also show that combining the two signals can further boost the performance to 80.5%, indicating that the two carry somewhat different but complementary information.
> We have updated our submission and now we only refer to this concept in the literature review section, and have clearly stated the source of this concept, in order to avoid any confusions about it.
>
> [1] Chen, Zhourong, et al. "You look twice: Gaternet for dynamic filter selection in cnns." Proceedings of the IEEE Conference on Computer Vision and Pattern Recognition. 2019.
>
> 2. Thank you for suggesting these baselines. We want to clarify that the concerns may be addressed by our existing experiments.
>     - For Baseline A, we feel that *training* BasisNet with fixed $\epsilon=1$ is equivalent to, therefore no better than, training one canonical MobileNet (because of static linear combination). However, we did validate the effectiveness of model synthesis during the *inference* stage. In Table 2 column “Uniform”, we show the experiments where bases are uniformly combined during inference, and the accuracy has significant degradation (-10%).
>     - For Baseline B, we emphasize that training cascaded MobileNets is equivalent with BasisNet with a single candidate, because the signals from the lightweight model are constant to be 1.0 after softmax activation. In Supplementary D.3, we show that when the BasisNet has only 1 candidate, the performance is the same as training a regular MobileNetV3-large (77.7%), indicating that the joint training of a regular cascade does not lead to the non-trivial performance boost of the BasisNet.
>
> We would like to thank you again for your time and valuable comments, and hopefully our explanations could address your concerns.

---

### Public Comment · ~Kane_Zhang1 · 2020-11-11
**Questions about the experiment**

Hi,

I found this to be a very interesting work. Compared with the previous CondConv and DynamicConv, BasisNet is easier to be deployed to different hardware accelerators on edge devices. It is because the coefficients are obtained from the lightweight model and the entire specialist model can be synthesized all at once.

I have two questions about the experiment.

Firstly, the teacher model is EfficientNet-B2 with noisy student training. Is it trained with the extra 300M unlabeled images?  In my understanding, it means your models are trained with the extra images indirectly while most of the compared models in Table 1 are not.

Secondly, I find in Appendix.B that your batch size is 16384 with 400K training steps. So, the number of epochs is 16384*4e5/1.28e6=5120. It is 10x larger than the one of normal experiment setting. I wonder whether this can bring a large improvement.

Thanks!

---

> ### Author Response · Authors · 2020-11-12
> **Thanks for your interest and we'd like to answer any other questions you may have.**
>
> Hi,
>
> We are glad to see that you find our paper interesting and helpful.
>
> Regarding your questions:
>
> (1) Thank you for pointing this out, and you are right that the EfficientNet teacher model we used is from noisy students training, which uses extra data. We'll update the notations. However, we want to emphasize that many of the listed methods in Table 1 (and Table 3 in Appendix C) use different training recipes, and we directly take the numbers from the corresponding original paper, thus it is just a rough comparison with literatures. We recommend to check our experiments and ablation sections, which have more careful, fair comparison.
> Specifically, in our experiments, we use the same teacher model checkpoint for training all models for both BasisNet and baselines (e.g., MobileNets and CondConv), thus the comparison is fair. Besides, we also experimented with using 1.4x MobileNetV2 as teacher model (see Table 4 in Appendix D.2), which is trained on ImageNet data only; the advantage of our BasisNet over MobileNet baseline is still quite significant (75.4% vs. 73.1%, under same training protocols).
>
> (2) Yes, our training protocol is slightly different since we used 8x8 TPU for training, and our re-implementation of baseline methods did show slightly better performance than reported in their original paper (e.g., MobileNetV2: reported 72.0%, we obtained 72.9%; CondConv-MV2: reported 74.6%, we obtained 76.2%). However, all experiments and ablations in our paper are performed under the exact same environment, with the same batch size, training epochs and any other conditions. Therefore we think the comparison is fair and our conclusion still holds, and the advantage of our proposed BasisNet is not coming from the prolonged training epochs.
>
> We are happy to answer any other questions you may have. Thanks for your interest!

---

### Author Response · Authors · 2020-11-24
**New experiments about on-device latency measurements to address practicality concern**

We thank all reviewers for their valuable suggestions, and we are glad to see that the merits of our methods are recognized by the reviewers.
Regarding the major concern about practicality, we performed experiments to measure the real latency of the proposed BasisNet on physical mobile device (Google Pixel 3XL) and the results are presented below (all models are floating-point models running on the big core of the phone’s CPU):

| Methods                    | Accuracy (%) | MAdds (M) | Latency (ms) |
|----------------------------|------------|---------|------------|
| BasisNet-MV3, 8 routes | 79.6 |281 | 60.6 |
| BasisNet-MV3, 16 routes    | 80.3      | 290     | 62.9   |
| -- With early stopping*     | 80.0       | 198     | 43.6    |
| |
| MobileNetV3-Large 1.25x224 | 79.7       | 356     | 66.3    |
| MobileNetV3-Large 1.5x224  | 80.6       | 489     | 86.2     |
| |
| CondConv-MobileNetV3       | 79.9     | 253     | 53.1    |

Note
*: The average latency with early stopping is estimated on ImageNet validation set. With a 0.7 threshold on ImageNet, ~39.3% of images are handled by the first stage (13.7ms) and the second stage is skipped; all images together still keep an accuracy of 80.0%. So the average latency is 39.3% * 13.7ms + (1 - 39.3% ) * 62.9ms = 43.6ms.

From the new experiments we show that:
1. BasisNet can run efficiently on existing mobile devices.  **Our efficiency conclusion drawn from MAdds also applies to real latency.**
2. MobileNetV3 with 1.25x and 1.5x multipliers have similar accuracy as BasisNet-MV3 with 8 and 16 routes, while the BasisNet has lower latency.
3. Our two-stage model synthesis design enables early stopping, which offers even more efficient execution (as shown by the estimated average latency of 43.6ms).
4. Comparison with CondConv:
    - CondConv has lower latency than BasisNet primarily because it does not use the first-stage lightweight model. However, the first stage computes basis weights with better results (80.3% vs 79.9%), and gets early stopping for free.
    - To more fairly compare with CondConv, one can add a first stage to CondConv but that may lead to (1) extra computation; (2) the overall accuracy still cannot go beyond 79.9% which is the accuracy of CondConv (i.e. upperbound when no image is skipped).
    - In Table 7 of Supplementary D.4, we show that combining CondConv and two-stage BasisNet can further boost performance to 80.5% (without early stopping), showing the BasisNet approach can actually work together with CondConv to combine the best of both.
    - BasisNet synthesizes a specialist that has the same architecture as existing state-of-the-art mobile networks. So it has the potential to be supported by any accelerators that are optimized for existing mobile network architectures. We leave exploring this direction as future work.


We hope the new experiments can address the concerns raised by reviewers about the practicality.

---

### Decision · Program_Chairs · 2021-01-07
**Final Decision**

**Decision:**

Reject

**Comment:**

The reviews were largely split in the beginning. Some of the concerns are firmly addressed, e.g. new results evaluating the actual latency in real hardware, and one reviewer raised the score from 5 to 6. However, another reviewer is not fully convinced by the response and decided to keep the original score of  "3: Clear rejection".

There are mainly two issues here. One is about the novelty of the method. The reviewer asked about the novelty and difference from CondConv/DynamicConv, however the authors emphasized the techniques to successfully train conditional computation models in general. As the focus of the paper is the proposal of the new (claimed as better) method, I have to say it is missing the points.  (The authors could have organized the storyline of the paper as "best practices for training conditional computation models" or something like that, if that is the true contribution the paper.) Another issue is about the advantage over the CondConv method. In the newly added results, BasisNet does not show clear advantage in terms of accuracy-speed trade-off without early exiting. Although the authors simply state that it is "infeasible" to do early termination on CondConv, the reason is not clear. Indeed, one can easily try layer-level early exiting as done in BranchNet for example, if not the model-level early exiting assumed for the BasisNet.

Base on the discussion above, I conclude that the paper has to clarify many issues before publication and thus recommend rejection.